

# A new genus of dance fly (Diptera: Empidoidea: Hybotidae) from Cretaceous Spanish ambers and introduction to the fossiliferous amber outcrop of La Hoya (Castellón Province, Spain)

Mónica M. Solórzano-Kraemer[1], Bradley J. Sinclair[2], Antonio Arillo[3] and Sergio Álvarez-Parra[4,5]

[1] Paläontologie und Historische Geologie, Senckenberg Forschungsinstitut und Naturmuseum, Frankfurt am Main, Germany
[2] Canadian Food Inspection Agency, Canadian National Collection of Insects, Ottawa, ON, Canada
[3] Departamento de Biodiversidad, Ecología y Evolución, Facultad de Biología, Universidad Complutense de Madrid, Madrid, Spain
[4] Departament de Dinàmica de la Terra i de l'Oceà, Facultat de Ciències de la Terra, Universitat de Barcelona, Barcelona, Barcelona, Spain
[5] Institut de Recerca de la Biodiversitat (IRBio), Universitat de Barcelona, Barcelona, Barcelona, Spain

Corresponding author
Mónica M. Solórzano-Kraemer,
monica.solorzano-kraemer@
senckenberg.de

## ABSTRACT

Hybotidae fly species, also known as dance flies, in Cretaceous ambers have been described from Lebanon, France, Myanmar, Russia, and Canada. Here we describe *Grimaldipeza coelica* **gen. et sp. n.**, and recognize another two un-named species, in Spanish amber from the middle Albian El Soplao and lower Cenomanian La Hoya outcrops. The fore tibial gland is present in the new genus, which is characteristic of the family Hybotidae. We compare *Grimaldipeza coelica* **gen. et sp. n.** with the holotypes of *Trichinites cretaceus* Hennig, 1970 and *Ecommocydromia difficilis* Schlüter, 1978, and clarify some morphological details present in the latter two species. Further taxonomic placement beyond family of the here described new genus was not possible and remains *incertae sedis* within Hybotidae until extant subfamilies are better defined. We provide new paleoecological data of the hybotids, together with paleogeographical and life paleoenvironmental notes. A table with the known Cretaceous Hybotidae is provided. Furthermore, the La Hoya amber-bearing outcrop is described in detail, filling the information gap for this deposit.

## INTRODUCTION

The Empidoidea Latreille, 1809 (Insecta: Diptera) contain more than 10,000 described species (*Pape, Blagoderov & Mostovski, 2011*), representing a diverse lineage within the 166,859 described species in the order Diptera (*Evenhuis & Pape, 2022*). The Empidoidea (dance flies and long-legged flies) consist of five families: Empididae Latreille, 1809;

Hybotidae Fallén, 1816; Atelestidae Henning, 1970; Dolichopodidae Latreille, 1809; and Brachystomatidae *sensu Sinclair & Cumming (2006)*. However, *Wahlberg & Johanson (2018)* returned the latter family to a lineage of the Empididae and elevated Ragadinae Sinclair, 2016 to family rank. A couple of additional families are sometimes also recognized in the Empidoidea (*Pape, Blagoderov & Mostovski, 2011*). The fossils studied here belong to the Hybotidae, a well-defined monophyletic group (*Sinclair & Cumming, 2006; Wahlberg & Johanson, 2018*) within the Empidoidea.

Most dance flies today are generalist predators, feeding on insects, but some also feed on dead insects (necrophagous). In addition, many species visit flowers and are known to feed on pollen and nectar (*Downes & Smith, 1969*). This behavior is also observable through the fossil record (*Grimaldi & Engel, 2005*). Predatory insects can be abundant in Defaunation resin, copal and amber because they are attracted by arthropods or vertebrates trapped by the resin (*Solórzano-Kraemer et al., 2015, 2018*).

The Hybotidae currently include seven subfamilies: Trichininae *Chvála, 1983*; Ocydromiinae Schiner, 1862; Oedaleinae *Chvála, 1983*; Tachydromiinae Meigen, 1822; Hybotinae Meigen, 1820; Stuckenbergomyiinae *Sinclair, 2019*; and Bicellariinae *Sinclair & Cumming, 2006* (*Sinclair & Cumming, 2006; Wahlberg & Johanson, 2018; Sinclair, 2019*). The subfamily Trichininae remains poorly defined and its relationships with the Oedaleinae, Ocydromiinae or Bicellariinae, or even its position within the Hybotidae, need to be further evaluated (*Sinclair & Cumming, 2006; Wahlberg & Johanson, 2018*).

The fossil record of the Cretaceous Hybotidae or unplaced hybotid-like species described in previous publications as *incertae sedis* is not extensive but diverse. Conversely, the Cenozoic fossils are abundant in Baltic, Dominican and Mexican ambers as well as compression fossils from the Oligocene of Brazil (EDNA Database, accessed November 2022). Cretaceous fossil hybotid species as bioinclusions have been described in amber from France, Lebanon, Myanmar, Russia, and Canada. Conversely, only one species based on a compression fossil is known from Orapa, Botswana (see Table 1). The species here described share characters with the genera *Trichinites Hennig, 1970* and *Ecommocydromia Schlüter, 1978*.

*Trichinites cretaceus Hennig, 1970* was described from Lebanese amber, which is Lower Cretaceous (Barremian, ~128 Ma) in age (*Maksoud & Azar, 2020*). It was described by *Hennig (1970)* based on a single female and placed as a stem-group to the subfamily group Ocydromioinea [Ocydromiinae+Hybotinae+Tachydromiinae], mainly based on wing characters. The position of the genus is currently assigned as the stem-group to the Hybotidae (*Chvála, 1983; Grimaldi & Cumming, 1999*). *Ecommocydromia difficilis Schlüter, 1978* was described in amber from Bezonnais, France, which is Cenomanian (~100 Ma) in age (*Schlüter, 1978; Perrichot et al., 2007*). The most important characters of this genus are the costa not circumambient, cell dm emitting three veins, and legs with pronounced setation. The fossil was placed within the Ocydromiinae, which at the time included all Hybotidae genera exclusive of Hybotinae and Tachydromiinae (*Schlüter, 1978*). Both Lebanese and French ambers were characterized within the group of *Agathis*-like (Araucariaceae) resins (*Perrichot et al., 2007; Azar, Gèze & Acra, 2010*)

**Table 1 Checklist of the known Cretaceous Hybotidae and related genera (Diptera: Empidoidea), with indication of the provenance and age.**

| Subfamily | Genus and species | Provenance | Age | Diagnosis | | | Reference |
|---|---|---|---|---|---|---|---|
| | | | | Head | Thorax | Abdomen | |
| Incertae sedis | **Grimaldipeza coelica gen. et sp. n.** (male) | El Soplao (Spain). Amber | Middle Albian (Early Cretaceous) (*García-Mondéjar, 1982*) | Proboscis half as long as head; eyes holoptic in males and dichoptic in females; antenna with long, two-articled arista-like stylus, longer than postpedicel. | Wing with $R_{2+3}$ straight to costa; cell dm slightly larger than cell cua; cell dm with $M_1$, $M_2$ and $M_4$ extending to wing margin. Thoracic setae long and strong. Fore tibial gland present. | Symmetrical male hypopygium, slightly rotated; epandrium with pair of articulated surstyli; left surstylus slightly elongate, with inner long, strong setae. | This article |
| Incertae sedis | **Grimaldipeza n. gen. sp. 1** (female) | El Soplao (Spain). Amber | Middle Albian (Early Cretaceous) (*García-Mondéjar, 1982*) | Proboscis 1/3 longer than head. | Bearing several shorter setae. Fore tibial gland present. | Bearing several shorter setae; abdomen seems to be not telescopic. | This article |
| Incertae sedis | *Grimaldipeza* n. gen. **sp. 2** (female) | La Hoya (Spain). Amber | Early Cenomanian (Late Cretaceous) (Barrón pers. Comm.) | Proboscis as long as the head. | With long and strong setae on the thorax and legs. Fore tibial gland present. | With long and strong setae; abdomen seems to be not telescopic. | This article |
| Incertae sedis | *Trichinites cretaceus Hennig, 1970* (female) | Jezzine (Lebanon). Amber | Barremian (Early Cretaceous) (*Maksoud & Azar, 2020*) | Proboscis short; postpedicel conical, with two-articled apical arista-like stylus. | Fore tibial gland absent or not visible. Wing with cell dm with $M_1$, $M_2$ and $M_4$ extending to wing margin, with distinct dm-cu vein; $R_{4+5}$ unforked; $R_{2+3}$ sharply curved prior to joining costa. | Bearing several shorter setae; telescopic abdomen. | Henning (1970); *Grimaldi & Cumming (1999)*, This article |
| Incertae sedis | *Ecommocydromia difficilis Schlüter, 1978* (male) | Écommoy (France). Amber | Cenomanian (Late Cretaceous) (*Schlüter, 1978*; *Perrichot et al., 2007*) | Postpedicel conical, with two-articled apical arista-like stylus. | Wing with cell dm with $M_1$, $M_2$ and $M_4$ extending to wing margin, with distinct dm-cu vein; $R_{4+5}$ unforked. Fore tibial gland appears present in abnormal position, but could be artifact due to preservation. | Hypopygium nearly symmetrical, not rotated; epandrium with left surstylus broader, not articulated; long postgonites or phallic process. | *Schlüter (1978)*, This article |

| Subfamily | Genus and species | Provenance | Age | Grounds to place the species in the subfamily | Reference |
|---|---|---|---|---|---|
| Hybotinae | *Pseudoacarterus orapaensis Waters, 1989* | Orapa (Botswana). Compression | Turonian (Late Cretaceous) (*Haggerty, Raber & Naeser, 1983*) | The species belongs to the subfamily because radial sector has only two branches and vein $R_{4+5}$ is unforked; wings more or less with developed axillary lobe; discal cell present, emitting 2 veins to wing margin; cell cua as long as basal cells; radial sector of intermediate length (*Waters, 1989*). | *Waters (1989)* |

(Continued)

| Subfamily | Genus and species | Provenance | Age | Grounds to place the species in the subfamily | Reference |
|---|---|---|---|---|---|
| Ocydromiinae | *Pouillonhybos venator Ngô-Muller et al., 2021* | Hukawng Valley (Myanmar). Amber | Early Cenomanian (Late Cretaceous) (*Shi et al., 2012*) | The species belongs to the subfamily because of cell cua is shorter than or about as long as cell bm, with outer angle obtuse; cell dm present; postpedicel shorter than arista-like stylus; proboscis oriented ventrally; epandrium with apical pair of articulated surstyli (*Ngô-Muller et al., 2021*). | *Ngô-Muller et al. (2021)* |
| Tachydromiinae | *Archiplatypalpus cretaceus Kovalev, 1974* | Yantardakh, Taimyr (Russia). Amber | Santonian (Late Cretaceous) (*Perkovsky & Vasilenko, 2019*) | These species belong to the subfamily because of the following characters: pterostigma absent; M2 absent; cell dm absent; hypandrium lacking apical lobes (*Sinclair & Cumming, 2006*). | *Kovalev (1974)*; *Grimaldi & Cumming (1999)* |
| | *Cretoplatypalpus americanus Grimaldi & Cumming, 1999* | Cedar Lake (Canada). Amber | Campanian (Late Cretaceous) | | *Grimaldi & Cumming (1999)* |
| | *Cretoplatypalpus archaeus Kovalev, 1978* | Nizhnyaya Agapa (Russia). Amber | Late Cenomanian (Late Cretaceous) (*Perkovsky & Vasilenko, 2019*) | | *Kovalev (1978)*; *Grimaldi & Cumming (1999)* |
| | *Electocyrtoma burmanica Cockerell, 1917* | Hukawng Valley (Myanmar). Amber | Early Cenomanian (Late Cretaceous) (*Shi et al., 2012*) | | *Cockerell (1917)*; *Grimaldi & Cumming (1999)* |
| | *Mesoplatypalpus carpenteri Grimaldi & Cumming, 1999* | Cedar Lake (Canada). Amber | Campanian (Late Cretaceous) (*McKellar & Wolfe, 2010*) | | *Grimaldi & Cumming (1999)* |

**Note:**
The new taxa here described are in bold. The genus *Cretoplatypalpus* is doubtfully assigned to Empidoidea *sensu Jouault et al. (2020)*. The genus *Ecommocydromia difficilis* was originally assigned to Ocydromiinae but later to the Empididae s.s. by *Grimaldi & Engel (2005)* and as *incertae sedis* within Empidoidea *sensu Ngô-Muller et al. (2021)* and in the present work. \*Trichinites has been proposed as the sister group of the hybotids, it is here included for practical reasons.

The specimens studied in this work come from the El Soplao and La Hoya amber-bearing outcrops (Fig. 1A), in Spain. The El Soplao outcrop is located in the western margin of the Basque-Cantabrian Basin (northern Iberian Peninsula). It belongs to the Las Peñosas Formation, dated as lower–middle Albian based on foraminifera (*García-Mondéjar, 1982*), and the amber is most probably middle Albian in age. The sedimentary environment is related to a delta-estuary under marine influence (*Najarro et al., 2009*). El Soplao amber has been extensively studied and is one of the richest in bioinclusions in the Iberian Peninsula, yielding a diverse arthropod fauna (*Najarro et al., 2010*). The La Hoya locality (not to be mistaken with the Barremian compression outcrop of Las Hoyas, also in Spain) has been mentioned in several congress communications and scientific publications (*Delclòs et al., 2007*; *Peñalver, Delclòs & Soriano, 2007*; *Peñalver et al., 2010*; *Peñalver & Delclòs, 2010*; *Menor-Salván et al., 2016*; *Murillo-Barroso et al., 2018*; *Rodrigo et al., 2018*; *McCoy et al., 2021*; *Santer et al., 2022*), but no detailed introduction about the general aspects of the outcrop has been published so far.

Here we describe more accurately the generalities of La Hoya outcrop. Furthermore, a new genus and species within Hybotidae are described and two additional, unnamed

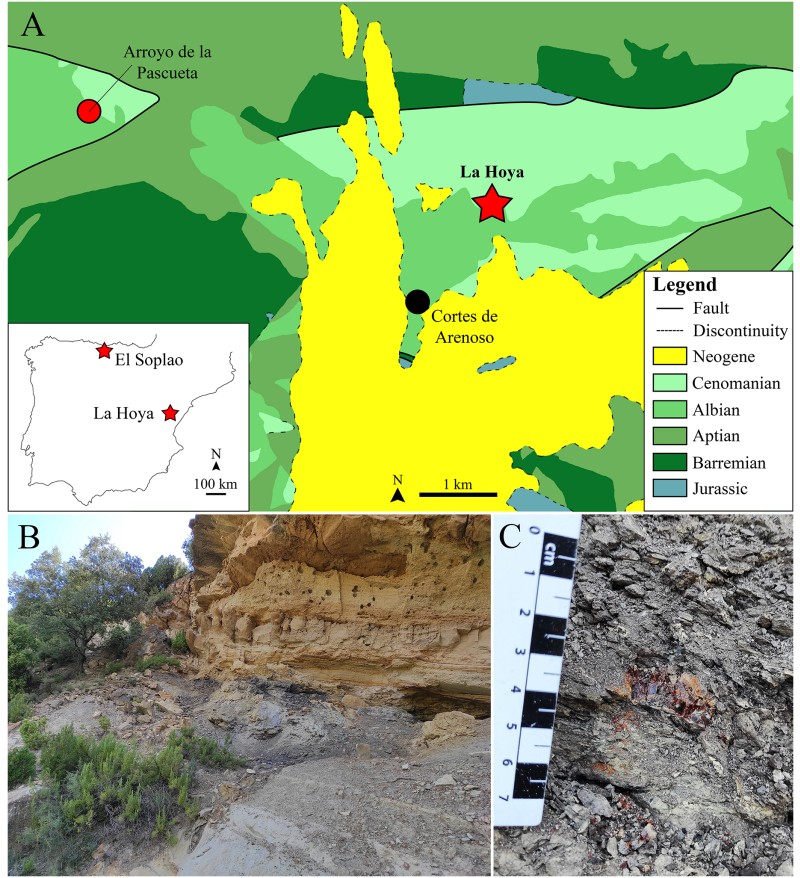

**Figure 1 The lower Cenomanian (Upper Cretaceous) amber-bearing outcrop of La Hoya (Maestrazgo Basin, Castellón Province, Spain).** (A) Geographical location within the Iberian Peninsula of the El Soplao (middle Albian) and La Hoya amber outcrops, and geological location of the La Hoya outcrop; Cortes de Arenoso village and the fossiliferous Arroyo de la Pascueta amber outcrop (late Albian) are also indicated in the geological map. (B) Amber-bearing level of the La Hoya outcrop, constituted by grey-black mudstone rich in organic matter, at the top of the Cortes de Arenoso section (E. Barrón, 2022, personal communication). (C) An amber piece in the rock of the La Hoya outcrop. Geological map in A modified and simplified from *Almera, Anadón & Godoy (1977)*.

species are recognized. We compare our specimens with the holotypes of *Trichinites cretaceus* and *Ecommocydromia difficilis*, and clarify and add some anatomical details of these latter species. Finally, we discuss the classification of these three genera within the Empidoidea.

## MATERIALS AND METHODS

Six specimens of dance flies included in amber have been examined for this work. From the El Soplao amber-bearing outcrop (Cantabria Autonomous Community, Spain): CES.404.1 ♂, CES.404.2 ♂, CES.439 ♂, CES.372 ♀; housed at the Colección Institucional del Laboratorio de la Cueva El Soplao in Celis, Cantabria (acronym for the collections is **CES**); permissions of excavations PFC 83/08 and PFC 33/09 (Consejería de Cultura, Turismo y Deporte del Gobierno de Cantabria, Spain), research agreement #20963 the

University of Barcelona. From the La Hoya amber-bearing outcrop (Castellón Province, Spain): MGUV-16348 (sex unknown) and MGUV-16349 ♀; housed at the Museu de la Universitat de València d'Història Natural (Burjassot, Valencia Province, Spain) (acronym for the collections is **MGUV**); permission of excavation 2003/0593-V (Conselleria d'Educació, Cultura i Esport de la Generalitat Valenciana, Spain). Amber pieces were cut and embedded in synthetic epoxy resin (EPO-TEK 301) and then polished (*Corral, López Del Valle & Alonso, 1999*; *Nascimbene & Silverstein, 2000*; *Sadowski et al., 2021*). Color photographs and Z-stack images were performed under a Nikon SMZ25 microscope, using Nikon SHR Plan Apo 0.5x and SHR Plan Apo 2x objectives with a Nikon DS-Ri2 camera and NIS-Element software (version 4.51.00; www.microscope.healthcare.nikon.com) and a digital camera attached to an Olympus BX51 compound microscope. Black and white infrared reflected photomicrographs were taken with a Nikon Eclipse ME600D (see *Brocke & Wilde, 2001* for precise technical information). Photographs were Z-stacked using the NIS-Element software. Drawings were made with the aid of an Olympus U-DA drawing tube attached to an Olympus BX50 compound microscope and digitized using a Wacom drawing tablet. Figures were assembled using Adobe Photoshop software (CS6 version 13.0; www.adobe.com).

The electronic version of this article in Portable Document Format (PDF) will represent a published work according to the International Commission on Zoological Nomenclature (ICZN), and hence the new names contained in the electronic version are effectively published under that Code from the electronic edition alone. This published work and the nomenclatural acts it contains have been registered in ZooBank, the online registration system for the ICZN. The ZooBank LSIDs (Life Science Identifiers) can be resolved and the associated information viewed through any standard web browser by appending the LSID to the prefix http://zoobank.org/. The LSID for this publication is: [LSID urn:lsid:zoobank.org:pub:D36ECF93-C05E-4A9C-8AA2-C799ED04346D]. The online version of this work is archived and available from the following digital repositories: PeerJ, PubMed Central SCIE and CLOCKSS.

Synchrotron Radiation micro-Computed Tomography (SRμ-CT) scans were carried out; however, the samples did not produce enough contrast, and segmentation of the reconstructed scans was not successful.

Following the methodology used previously by our research group (*e.g.*, *Álvarez-Parra et al., 2021*; *Sarto i Monteys et al., 2022*), the Fourier Transform Infrared Spectroscopy (FTIR) analysis of La Hoya amber was obtained through an IR PerkinElmer Frontier spectrometer using a diamond ATR system with a temperature stabilized DTGS detector and a CsI beam splitter at the Molecular Spectrometry Unit of the CCiTUB (University of Barcelona, Spain).

The anatomical terminology of the specimens follows *Cumming & Wood (2017)*.

# RESULTS

### Systematic paleontology

The specimens of the species described here are placed in the family Hybotidae, based on the following apomorphic ground plan characters: (l) Costa runs to just below the apex

of the wing, ending near or beyond distal end of $M_1$ or $M_{1+2}$; (2) Sc incomplete, not reaching the wing margin, ending freely in wing membrane; (3) $R_{4+5}$ unbranched; (4) fore tibial gland present; (5) stylus with bare, terminal sensillum.

Order DIPTERA Linnaeus, 1758
Superfamily Empidoidea (*sensu* Chvála, 1983)
Family HYBOTIDAE Meigen, 1820

*Grimaldipeza* **n. gen.** (Figs. 2–7)
LSID urn:lsid:zoobank.org:act:23818B66-56D4-4D7B-BFA5-1EB7D6995462

**Type species.** *Grimaldipeza coelica* **n. sp.**

**Etymology.** The genus name is in honour of David A. Grimaldi (American Museum of Natural History, New York, NY, USA) for his persevering and remarkable legacy work on fossil insects, principally on Diptera and the common Greek suffix in Empidoids *peza*, meaning foot. The gender is feminine.

**Diagnosis.** Eyes meeting above antennae (holoptic) in males and dichoptic in females. Antenna with long, arista-like stylus, longer than postpedicel. $R_{2+3}$ straight to costa. Cell dm slightly larger than cell cua, apices in same plane or linear. Cell dm with $M_1$, $M_2$ and $M_4$ extending to wing margin. Thoracic setae long and strong. Fore and hind tibiae more or less of the same thickness. Longitudinal furrow on mid and hind femora and tibiae. Fore tibial gland present. Symmetrical male hypopygium, slightly rotated. Hypandrium apically narrowly bilobed, with posterior apices pointed.

*Grimaldipeza coelica* **n. sp.** (Figs. 2–4, and 5A–5C)
LSID urn:lsid:zoobank.org:act:9028DF70-F0B3-4A2C-B677-64F51C8CEC33

**Etymology.** After the locality of Celis, close to the outcrop El Soplao in Cantabria Autonomous Community, Spain.

**Diagnosis.** As for the genus.
HOLOTYPE: **CES.404.1** ♂. Housed at the Colección Institucional del Laboratorio de la Cueva El Soplao in Celis (Cantabria, SPAIN).
PARATYPES: **CES.404.2** ♂, **CES.439** ♂

**Description**
**Body.** Holotype male CES.404.1 (Figs. 2 and 3) body length about 1.79 mm, wing length 1.57 mm. Paratype male CES.404.2 (Fig. 2A right) body length 1.74 mm, wing length 1.53 mm. Paratype male CES.439 (Figs. 4A–4E) body length 1.78 mm, wing length 1.42 mm.

**Head.** Eyes meeting above antennae (holoptic) (Fig. 3B); eyes do not meet below antenna (Fig. 3A); ommatrichia absent. Face flat level with eyes. Gena not extended below eye; ventral surface of head, posterior to mouth opening clothed in long, pale setulae. Two pairs of fine ocellar setulae, directed more upward than forward. One pair of outer vertical setae

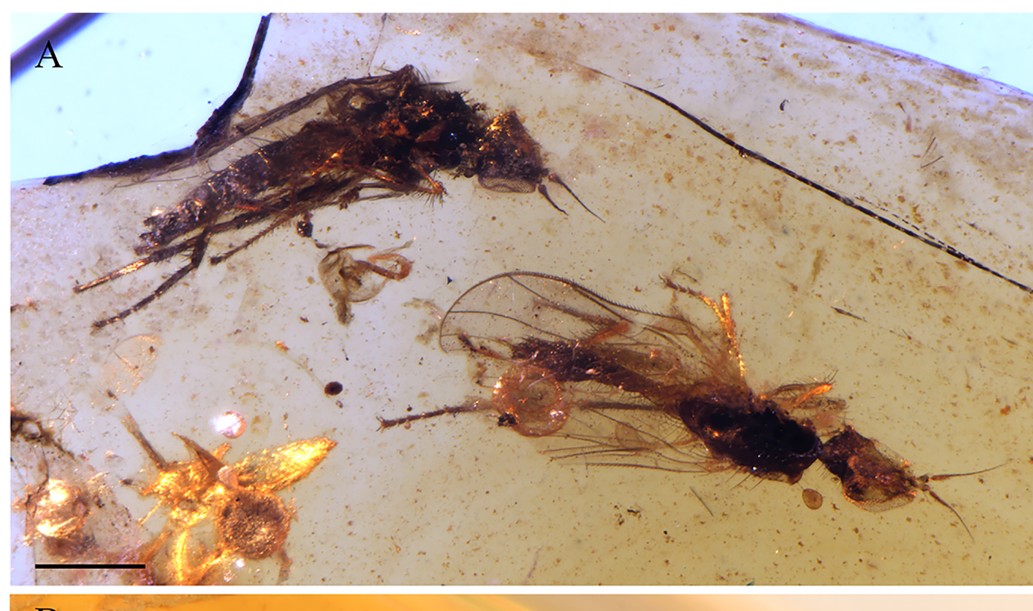

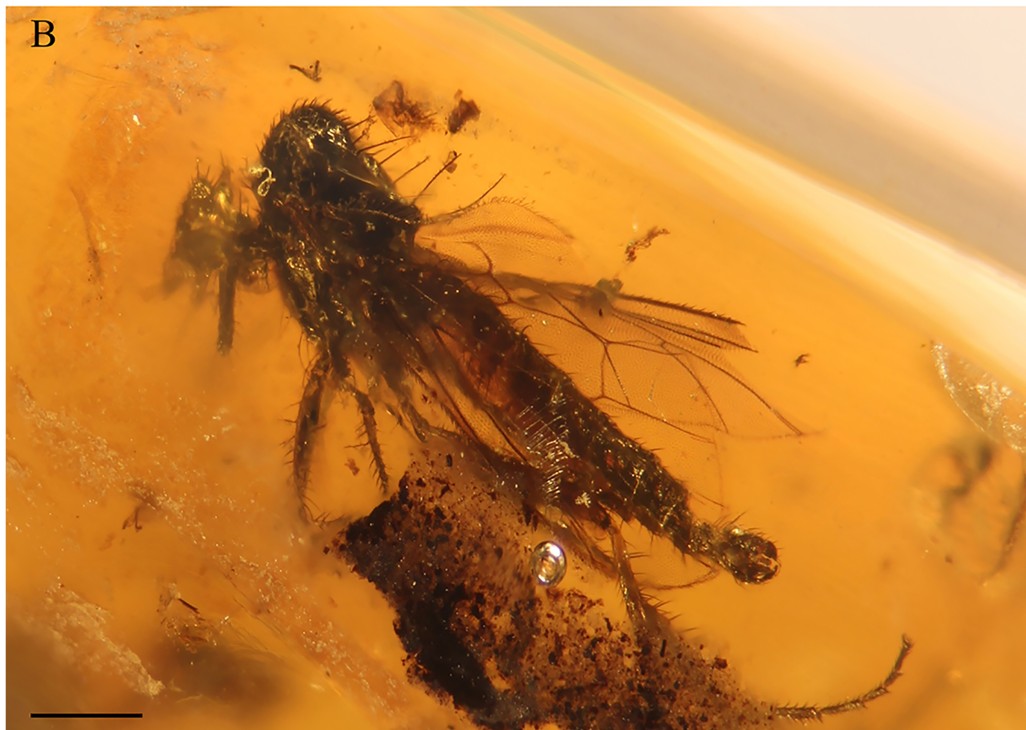

**Figure 2** *Grimaldipeza coelica* n. gen. et sp. (Diptera: Hybotidae) , holotype male CES.404.1 and paratype male CES.404.2, from El Soplao outcrop, Cantabria, Spain (middle Albian in age). (A) Piece showing two specimens (holotype, left; paratype, right). (B) Habitus of holotype CES.404.1, lateral view. (C) Habitus of holotype CES.404.1, dorsal view. Scale bars 0.5 mm.

and one pair of inner vertical setae. Antenna inserted above middle of head; scape small, devoid of setulae; pedicel globose bearing long setulae; scape and pedicel similar in length; postpedicel pointed ovate to conical, apically tapered gradually to point, with two-articled apical arista-like stylus (Figs. 3A–3C), longer than postpedicel; stylus 0.22 mm, pubescent with bare, terminal sensillum. Labrum large, almost as long as proboscis, apex of labrum

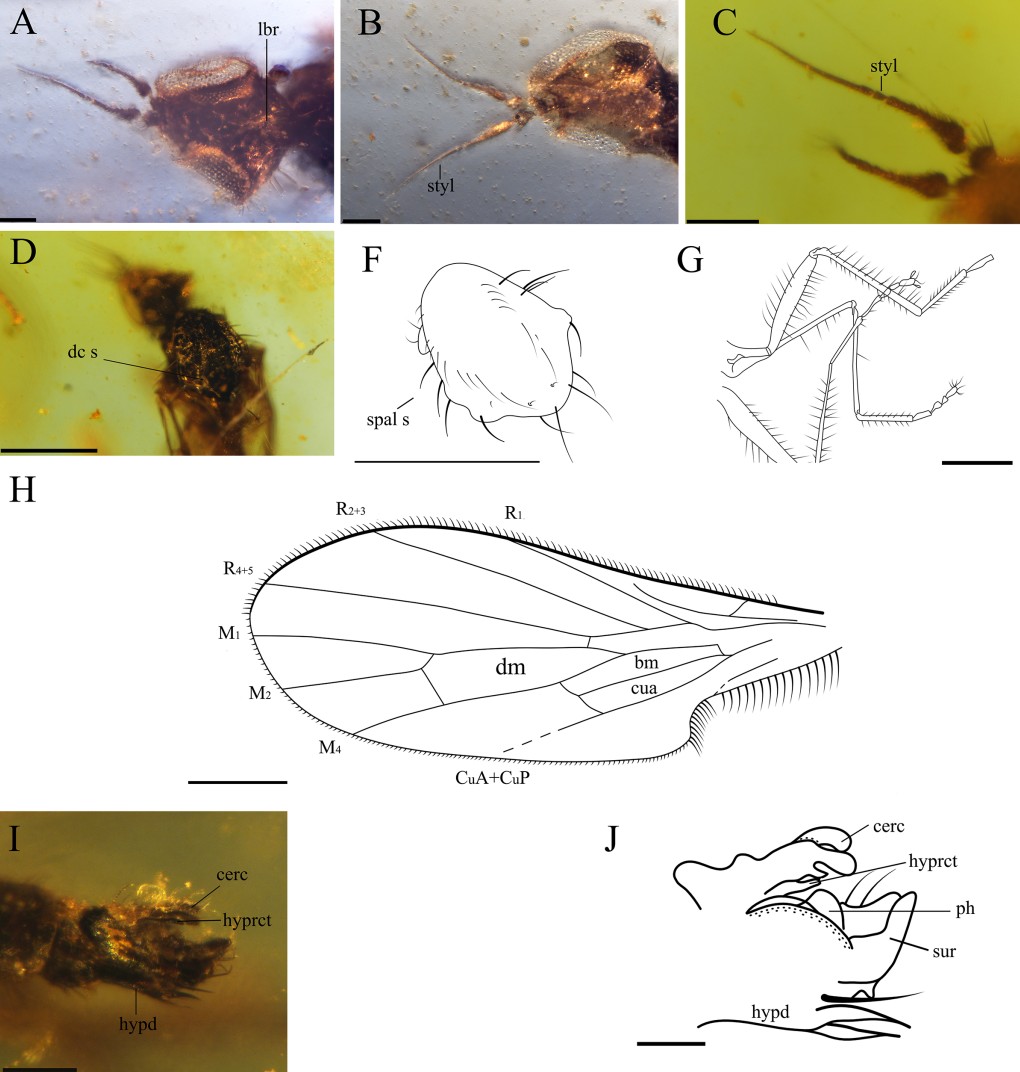

**Figure 3 Important characters of *Grimaldipeza coelica* n. gen. et sp. (Diptera: Hybotidae), holotype male CES.404.1 and paratype male CES.404.2, from El Soplao outcrop, Cantabria, Spain (middle Albian in age).** (A) Head of holotype CES.404.1, ventral view. (B) Head of paratype male, dorsal view. (C) Antennae of holotype. (D) Thorax of holotype, dorsal view. (F) Drawing of thorax of holotype, dorsal view. (G) Hindlegs of holotype. (H) Male terminalia of holotype, lateral view. (I) Male terminalia of holotype, Ia laterodorsal right, Ib lateroventral left. (J) Male terminalia of paratype male. Scale bars (A–C) and (H–J), 0.1 mm, (D–G) 0.5 mm. Abbreviations: cerc, cercus; dc s, dorsocentral setae; hypd, hypandrium; hyprct, hypoproct; lbr, labrum; ph, phallus; spal s, supra-alar seta; styl, stylus; sur, surstylus.

rounded. Proboscis half as long as head. Palpus round, bearing 3–4 long setae; palpifer not visible or not present. Labellum covered with short setae on base and four strong, short setae visible in specimen CES.439 (Figs. 4B and 4C).

**Thorax.** Notum humpbacked (Fig. 2B), notum with four irregular rows of acrostichal and dorsocentral setulae; two pairs of larger dorsocentral setae posteriorly. One long, strong supra-alar seta, two long, strong notopleural setae, and one postalar seta on each side

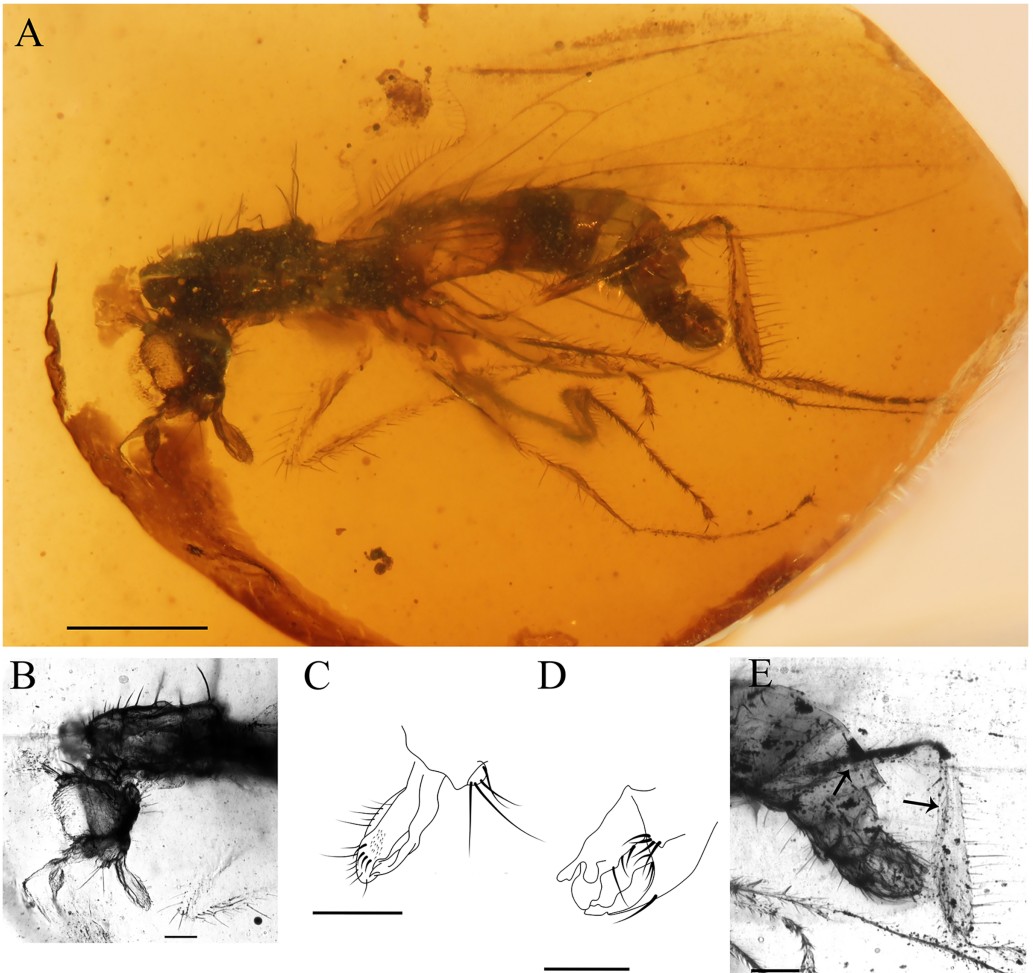

**Figure 4** *Grimaldipeza coelica* **n. gen. et sp. (Diptera: Hybotidae), paratype male CES.439, from El Soplao outcrop, Cantabria, Spain (middle Albian in age).** (A) Habitus, lateral view. (B) Drawing of habitus, lateral view. (C) Head, ventrolateral view. (D) Mouthparts. (E) Male terminalia, laterodorsal view. (F) Photo of male terminalia, laterodorsal view and hindleg. Black arrows show the longitudinal furrow on hind femur and tibia. (G) Reconstruction of wing. Scale bars (A, B and G) 0.5 mm, (C–F) 0.1 mm. Abbreviations: bm, basal medial cell; cua, anterior cubital cell; CuA+CuP, anterior branch of cubital vein + posterior branch of cubital vein; dm, discal medial cell; M, medial vein; R, radial vein; Sc, subcostal.

(Fig. 3F). Scutellum with two pairs of long setae. Legs long and unmodified, hindlegs longest; none of legs raptorial. Fore and hind femora slightly thicker than mid femur. Mid and hind femora and tibiae with longitudinal furrow (Fig. 4F). All femora and tibiae armed with rows of long, strong setae (Figs. 2A and 3G). Tarsus of all legs bearing short, strong setae. Fore tibia with posteroventral gland (Fig. 5).

**Wing.** Hyaline; with fine microtrichia over entire membrane. Pterostigma absent. Costa terminates between $R_{4+5}$ and $M_1$ (Fig. 3H); Sc apically evanescent, ending slightly before costal margin; Rs arising distant from level of humeral crossvein; $R_1$ ending at or slightly beyond mid-length of wing; $R_{2+3}$ straight to C, ending closer to apex of $R_1$ than $R_{4+5}$; $R_{4+5}$ unbranched, parallel to $M_1$; cell dm slightly larger than cell cua, emitting three veins: $M_1$,

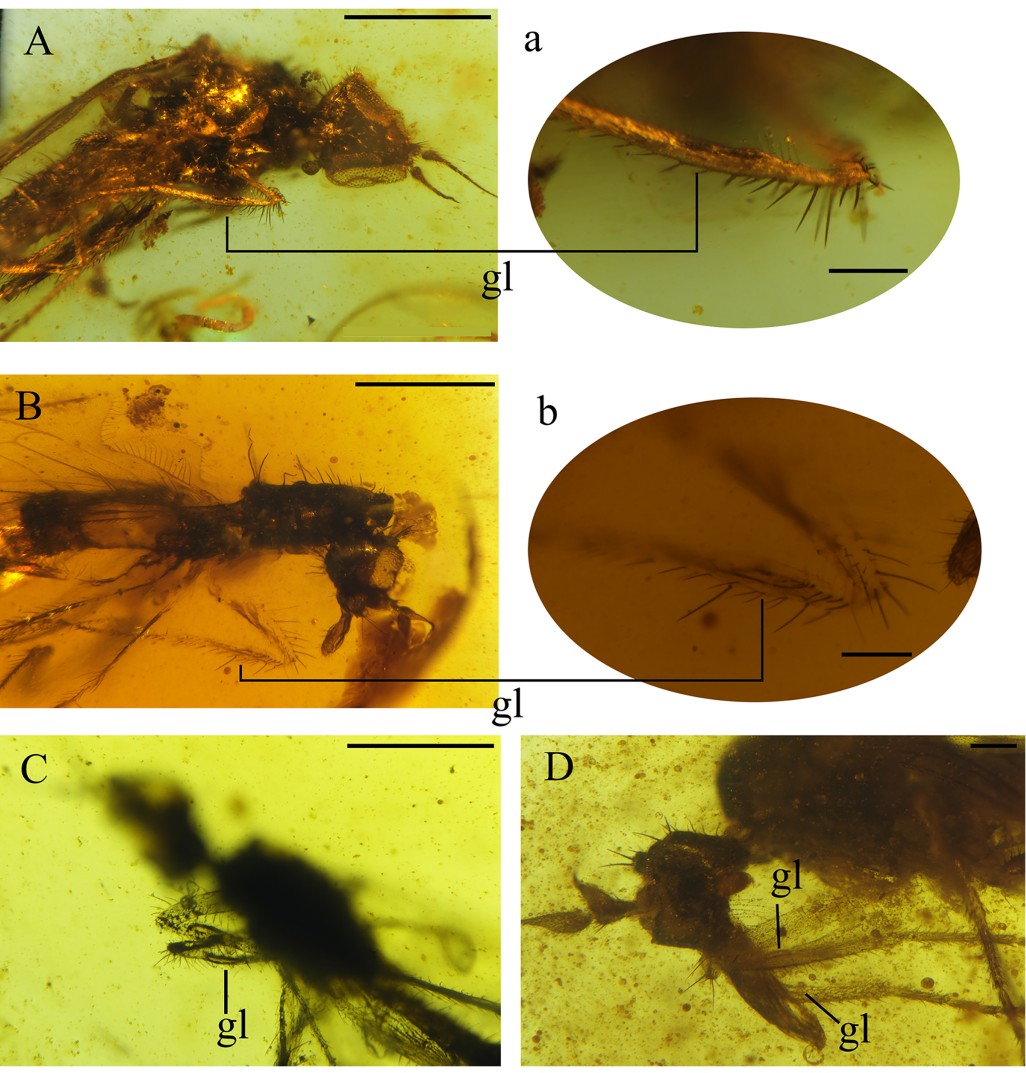

**Figure 5 Fore tibial gland in *Grimaldipeza coelica* n. gen. et sp. (Diptera: Hybotidae), from El Soplao outcrop, Cantabria, Spain (middle Albian in age).** (A) Holotype CES.404.1, with close-up of gland. (B) Paratype CES.439, with close-up of gland. (C) Paratype CES.404.1. (D) Female, CES.372. Scale bars (A–C) 0.5 mm, (a, b and D) 0.1 mm. Abbreviation: gl, gland.

$M_2$, and $M_4$; $M_1$ and $M_4$ only moderately divergent; CuA straight, aligned with apex of cell bm; apex of cell cua slightly truncate or acute. Anal lobe broad, well developed.

**Abdomen.** Abdomen scarcely broader near base, laterally compressed. Tergites and sternites bearing long, strong setae. Hypopygium symmetrical, slightly rotated (Figs. 3I–3J, 4D and 4E). Hypandrium apically narrowly bilobed, with posterior apices pointed (Figs. 3I–3J). Epandrium with pair of articulated surstyli; left surstylus slightly elongate, with inner long, strong setae. Cercus short, unmodified with pubescence.

*Grimaldipeza* species 1

(Fig. 6)

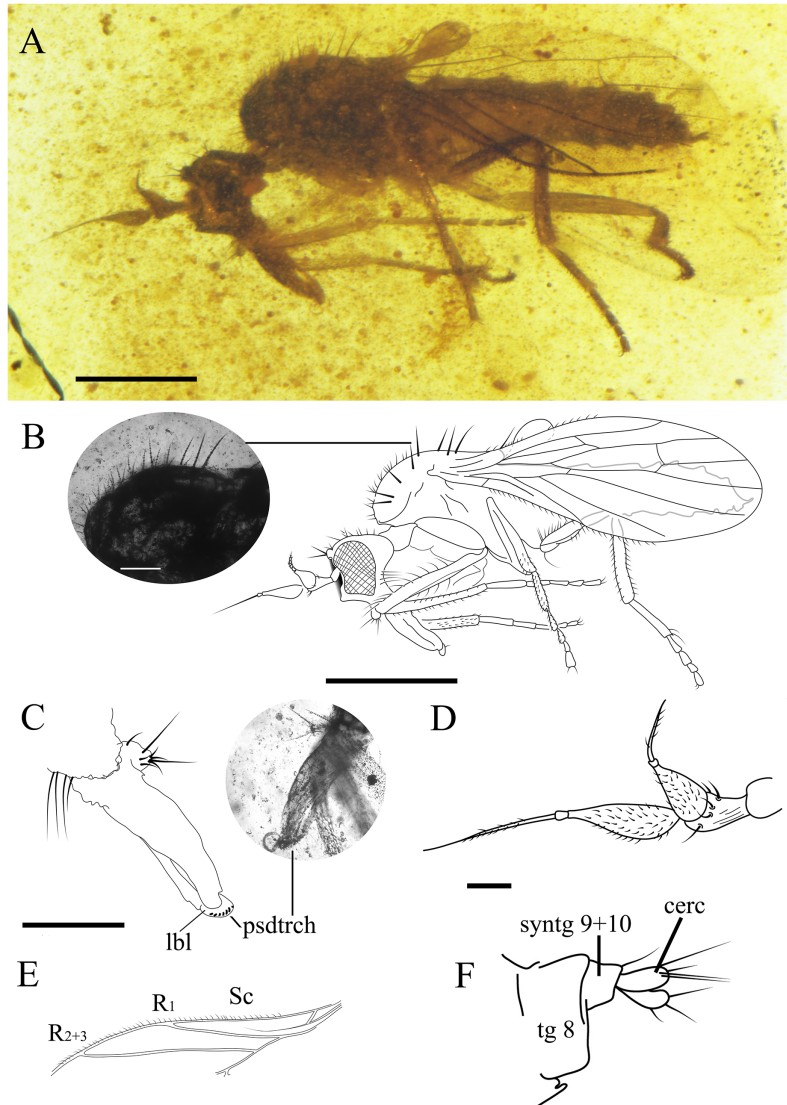

**Figure 6 Female *Grimaldipeza* sp. 1. (Diptera: Hybotidae), female CES.372, from El Soplao outcrop, Cantabria, Spain (middle Albian in age).** (A) Habitus. (B) Drawing of habitus with photo showing thoracic setae with apparent seriated rings (scale bar 0.1 mm). (C) Mouthparts. (D) Antennae. (E) Anterior part of wing. (F) Terminalia. Scale bars (A, B, and E) 0.5 mm; (C, D and F) 0.1 mm. Abbreviations: cerc, cercus; lbl, labrum; psdtrch, pseudotrachea; R, radial vein; Sc, subcostal; tg, tergite.

**Female. CES.372 ♀.** Body length 1.68 mm. Wing 1.22 mm. Stylus 0.22 mm. Postpedicel 0.16 mm. Similar to *Grimaldipeza coelica* gen. et sp. n. male holotype, except for the following characters: eyes dichoptic (Figs. 6A and 6B); setae shorter and finer on thorax, abdomen and legs; labrum curved bearing pseudotracheae (Fig. 6C); proboscis 1/3 longer than head; thorax setae appears with seriated rings (Fig. 6B), possibly artifact of preservation; tergites bearing several shorter setae in comparison with males, first tergite with cluster of about five long setae on each side; apical abdominal segments exposed, gradually telescopic; cercus cylindrical, bearing three long setulae (Fig. 6F).

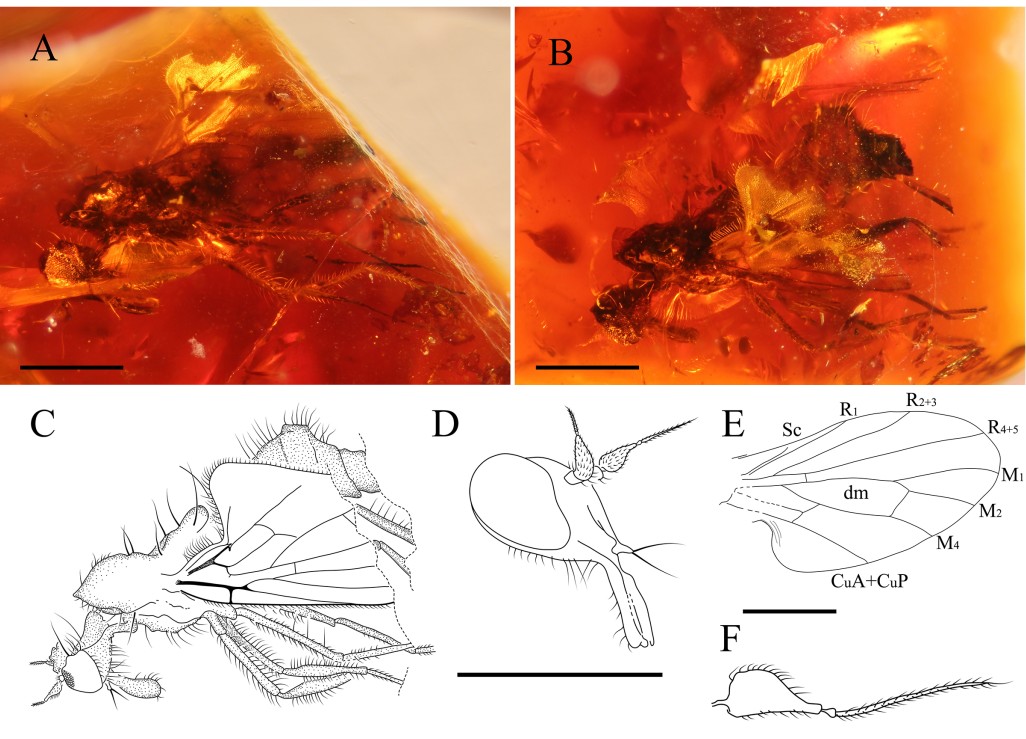

**Figure 7 *Grimaldipeza* sp. 2. (Diptera: Hybotidae), MGUV-16348 (sex unknown), from La Hoya outcrop, Castellón, Spain (early Cenomanian in age).** (A) Habitus, lateral view right. (B) Habitus, lateral view left. (C) Drawing of habitus, lateral view. (D) Head, laterofrontal view. (E) Wing reconstruction. (F) Antenna. Scale bars (A–E) 0.5 mm; (F) 0.1 mm. Abbreviations: CuA+CuP, anterior branch of cubital vein + posterior branch of cubital vein; dm, discal medial cell; M, medial vein; R, radial vein; Sc, subcostal.

**Remarks.** *Grimaldipeza* sp. 1 can be distinguished from *Grimaldipeza* sp. 2 by the tergites bearing several shorter setae and the proboscis 1/3 longer than head.

### *Grimaldipeza* species 2
(Fig. 7)

**Female. MGUV-16348** (sex unknown), **MGUV-16349** ♀. Part of the abdomen of MGUV-16348 is not preserved; however, all other characters, such as the length of the proboscis, which is as long as the head, and dichoptic eyes indicate that it could be a female. Specimen MGUV-16349 only shows the end of the abdomen, which is telescopic, and one wing partially and badly preserved. MGUV-16348 and MGUV-16349 are included in the same amber piece as syninclusions. Similar to the male holotype of *Grimaldipeza coelica* gen. et sp. n. except for the following characters: stylus 0.19 mm; postpedicel 0.10 mm; eyes dichoptic (Fig. 7D); setae long, strong on thorax, abdomen and legs, somewhat longer than in *Grimaldipeza* sp. 1; labrum curved; proboscis 1/4 longer than head (Fig. 7D); tergites and sternites bearing several long setae; setae on anal lobe of wing longer than in

*Grimaldipeza* sp. 1; apical abdominal segments exposed, gradually telescopic; cercus short, cylindrical.

**Remarks.** *Grimaldipeza* sp. 2 can be distinguished from *Grimaldipeza* sp. 1 by the long and strong setae on the thorax, abdomen and legs, somewhat longer than in species 1 and the proboscis is as long as the head. As the females do not appear in any of the pieces containing males we cannot describe these as new species and they will remain unnamed until more specimens are found, that can be associated to the species with confidence.

 *Grimaldipeza* n. gen. can be distinguished from *Trichinites* by the following combination of characters: position of vein r-m close to base of cell dm, $R_{2+3}$ longer than in *Trichinites* and extending straight to wing margin (*T. cretaceus* is sharply curved prior to joining costa, Fig. 8A). Apices of cell bm and cua are aligned in *Grimaldipeza* n. gen. In contrast, the apex of cell cua is obliquely projecting in *Trichinites*. *Trichinites* has an extra cell at the bifurcation of $M_1$ and $M_2$, but this is most probably an aberrant feature, thus it is not a diagnostic feature of the species, and is absent in all other specimens studied here. In the new genus, the fore tibia is slightly broader than the mid and hind tibiae and the fore tibial gland is present, distinguishable in three of the four specimens (Fig. 5). Furthermore, *Trichinites* lacks the fore tibial gland and is larger than the new species here described (body length 2.99 mm), including the telescopic abdomen, whereas the maximum length of *Grimaldipeza coelica* gen. et sp. n. is 1.79 mm. The thorax of the paratype CES.439 appears somewhat flat, however, this is not considered here a differential character and could be due to the fossilization process.

*Ecommocydromia difficilis* Schlüter, 1978
(Fig. 9)

**Complementary description**

**Head.** Antenna inserted above middle of head. Right antenna broken, only scape, pedicel and part of postpedicel preserved. Left antenna preserved but base not visible. Scape longer than pedicel; pedicel short, slightly broader than scape with one long seta visible; postpedicel on one side appears pointed ovate to conical, however (as described by *Schlüter (1978)*) left postpedicel conical, with two-articled apical arista-like stylus (Figs. 9E and 9F).

**Thorax.** Laterotergite bare. Presence of tibial gland on foreleg not possible to ascertain. On right fore tibia, gland appears present, in abnormal position (Fig. 9H), but could be artifact due to preservation. On left foreleg, gland not visible, or absent. Fore tibia with anterodorsal row of strong setae, length nearly as long as width of tibia (Fig. 9G).

**Abdomen.** Shorter than thorax. Tergites and sternites bearing long, strong setae. Hypopygium nearly symmetrical, not rotated (Figs. 9C and 9D). Hypandrium apically narrowly bilobed, with posterior apices pointed (Fig. 9D). Epandrium with left surstylus broader, not articulated; long postgonites or phallic process (Fig. 9D). Cercus short, unmodified.

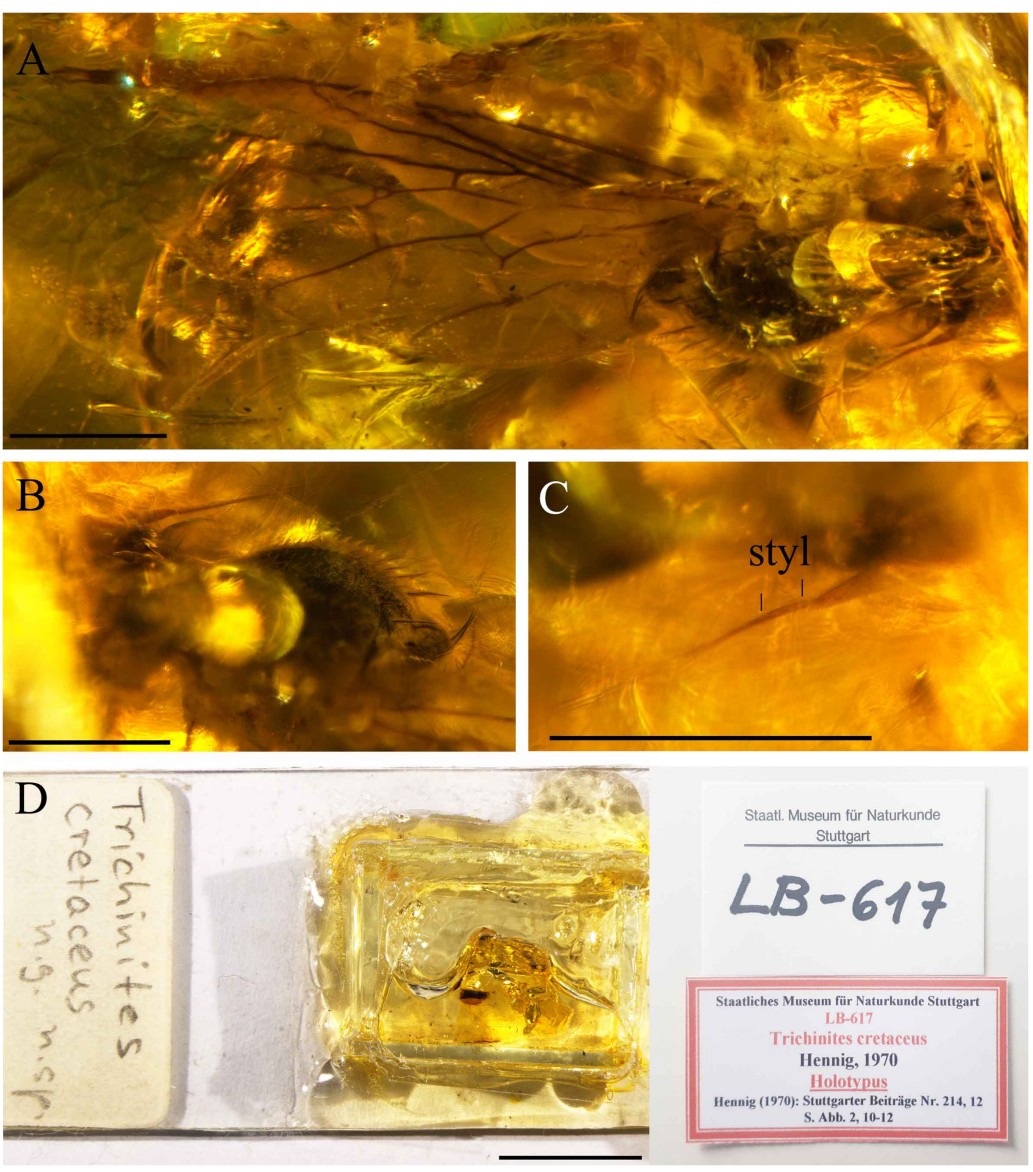

**Figure 8** *Trichinites cretaceus Hennig, 1970*, **holotype Number LB-617, from Jezzine outcrop, Lebanon (Barremian in age).** (A) Scutum and wing, dorsal view. (B) Thorax, oblique lateral view. (C) Antenna with one basal article. (D) Holotype with labels. Scale bars (A–C) 0.5 mm, (D) 10 mm. Abbreviation: styl, stylus.

**Remarks.** In the original publication, the holotype of *E. difficilis* has the collection number Emp Ce Bez 1 (Paläontologisches Institut, FU-Berlin) (Fig. 9I). However, the holotype has been transferred to the Natur Museum für Naturkunde in Berlin, Germany under the Number MB.I.7927.

**La Hoya amber-bearing outcrop**

The La Hoya amber-bearing outcrop is located in the Penyagolosa Sub-basin within the Maestrazgo Basin in the eastern Iberian Peninsula (*Salas & Guimerà, 1996*). More than 30 amber outcrops have been reported in this basin, although only four of them are

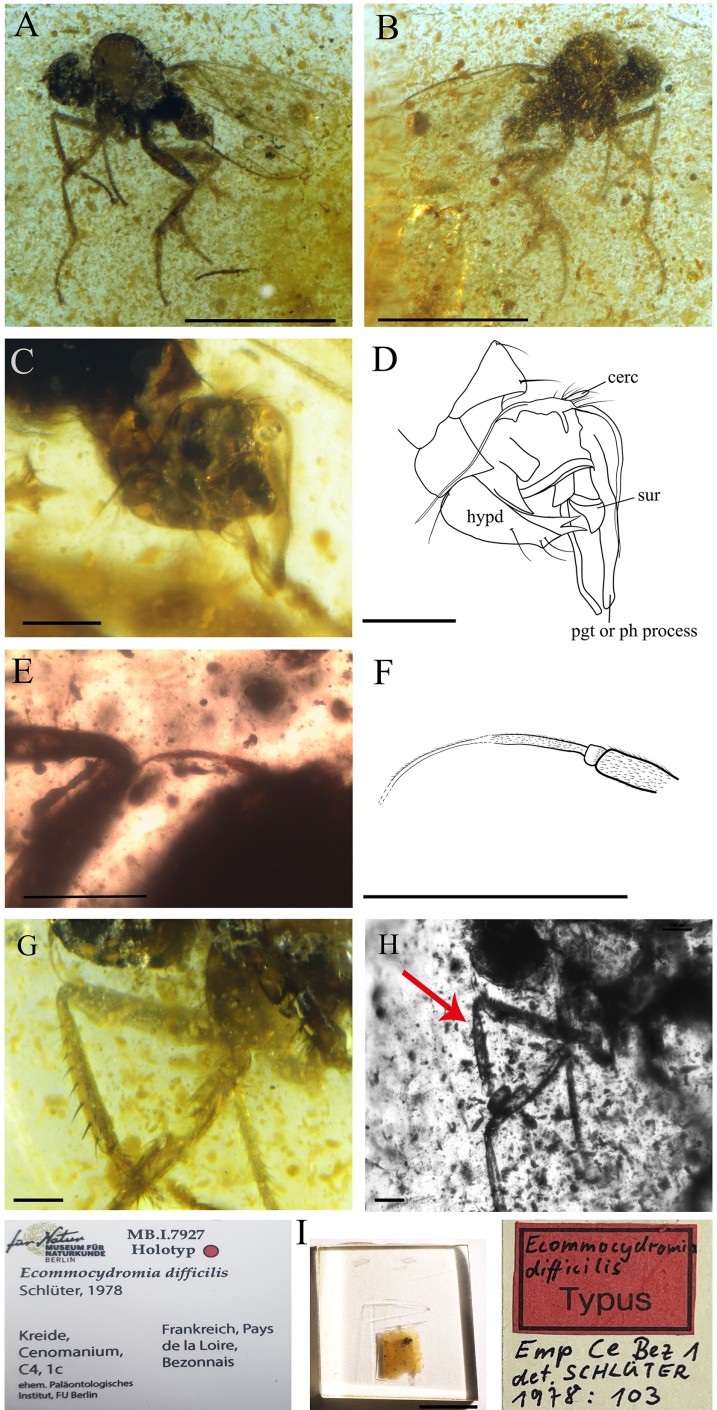

**Figure 9 *Ecommocydromia difficilis Schlüter, 1978*, from Écommoy, France (Cenomanian in age).**
(A) Habitus in left lateral view. (B) Habitus in right lateral view. (C) Terminalia in left lateral view.
(D) Drawing of terminalia in left lateral view. (E) Antenna. (F) Drawing of antenna. (G) Foreleg.
(H) Forelegs photographed with infrared camera, red arrow indicates possible posteroventral gland,
however it could also be an artifact. (I) Holotype with labels. Scale bars (A and B) 0.5 mm, (C–H) 0.1 mm,
(I) 5 mm. Abbreviations: cerc, cercus; hypd, hypandrium; pgt, postgonite; ph process, phallic process; sur,
surstylus.                                                           

fossiliferous (*Peñalver & Delclòs, 2010*; *Álvarez-Parra et al., 2021*): Ariño, San Just, Arroyo de la Pascueta and La Hoya. The amber outcrop of La Hoya is close to the Cortes de Arenoso town (Castellón Province, Valencian Community) and was named after the Font de l'Hoya ravine, where it is located. The Arroyo de la Pascueta amber outcrop is only a few kilometers from La Hoya (Fig. 1A). The oldest mention of amber in the Valencian Community corresponds to *Cavanilles (1797)*, who indicated the presence of "succino" near the Quesa town (Valencia Province); since then, the number of amber outcrops detected in the region has increased. The amber of this area was traditionally used as incense by shepherds, but the livestock grazing has declined in the last decades, furthermore the access to the outcrop is difficult, so this locality could be currently free of anthropic alteration (*Rodrigo et al., 2018*). The La Hoya amber outcrop was discovered in 1998 and the first paleontological excavation took place in October 2003. La Hoya amber corresponds to the only known fossiliferous amber from the Valencian Community, including: two cockroaches (Blattodea), one platygastrid (Hymenoptera), one chironomid (Diptera), two hybotids (Diptera) here studied, and a few undetermined insect remains.

Geologically, the La Hoya amber outcrop is located at the top of the Cortes de Arenoso section (E. Barrón, 2022, personal communication). This section has been dated as upper Albian–lower Cenomanian based on stratigraphic and palynological data, while the amber-bearing level is most probably lower Cenomanian (Upper Cretaceous) (E. Barrón, 2022, personal communication). Therefore, La Hoya corresponds to the only known fossiliferous Cenomanian amber outcrop from the Iberian Peninsula, providing an interesting comparison framework with the Albian fossiliferous ambers from Iberia and with other Cenomanian ambers, such as those from the Hukawng Valley (Myanmar) and Charente-Maritime (France) (*Grimaldi, Engel & Nascimbene, 2002*; *Perrichot et al., 2007*). The amber outcrops of San Just, Arroyo de la Pascueta, and La Hoya were initially assigned to the Escucha Formation (*Delclòs et al., 2007*), but they actually correspond to the Utrillas Group (E. Barrón, 2022, personal communication). The amber-bearing level of La Hoya is a grey-black mudstone rich in organic matter about 50 cm thick at the top of grey mudstone about three meters thick (Fig. 1B). Below the grey mudstone there is a sandstone level, while above the amber-bearing level there is a limestone level (Fig. 1B). The amber-bearing rock is tough, and the amber pieces are usually broken and crumbled, so amber extraction is challenging (Fig. 1C). The aerial amber pieces (related to resin produced in branches or trunks) are scarcer than the nearly rounded kidney-shaped pieces (related to resin produced in roots). The aerial amber mainly corresponds to flow-shaped pieces, instead of droplet- or stalactitic-shaped morphologies. The color of the amber pieces is reddish-yellow.

The FTIR spectrum of the aerial amber from the La Hoya outcrop (Fig. 10) shows the typical characteristics of the amber (*Grimalt et al., 1988*): carbon-hydrogen stretching band about 2,950 cm$^{-1}$, a prominent carbonyl band about 1,700 cm$^{-1}$, and bending motions carbon-hydrogen bands about 1,470 and 1,380 cm$^{-1}$. There are also hydroxyl bands about 3,500 cm$^{-1}$. The absence of exocyclic methylenic bands at 1,640 and 880 cm$^{-1}$ indicates a high degree of maturation, related to the Cretaceous age of the amber (*Alonso et al., 2000*). The molecular composition of the La Hoya amber (through gas chromatography-mass

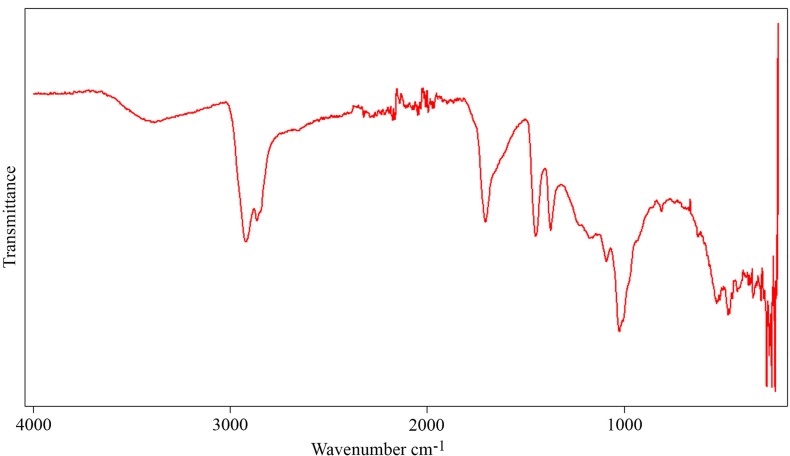

**Figure 10 Infrared spectroscopy spectrum (FTIR) from an aerial amber piece of La Hoya outcrop (Castellón Province, Spain).** Resolution = 4 cm$^{-1}$.  

spectrometry) was classified as Type 3, based on an absence of abietane type diterpenoids and higher proportion of amberene with relatively lower homoamberene (**III**) and trimethyltetralin (**II**) than the Type 1 amber (*Menor-Salván et al., 2016*). The amber Type 3 is compatible with an Araucariaceae origin (*Menor-Salván et al., 2016*; *McCoy et al., 2021*).

Finally, it is important to note that the La Hoya amber-bearing outcrop is designated as a LIG (*Lugar de Interés Geológico*, Site of Geological Interest), and it is protected under legislation for paleontological heritage, which means that the excavation requires previous permission from the regional government and that the extracted samples should be deposited in a public institution within the region (*Rodrigo et al., 2018*).

## DISCUSSION

### Systematic position

The description of *Trichinites cretaceus* was based on a single female in Barremian amber from Jezzine (Lebanon) and is housed at the Staatliches Museum für Naturkunde (Stuttgart, Germany) (Holotype Nummer LB-617) (Fig. 8D). This holotype specimen was re-examined. *Trichinites* has been proposed as the sister group of the hybotids in which the fore tibial gland was not developed (*Chvála, 1983*) and was described with the following characters: arista-like stylus longer than postpedicel, with two, possibly three articles (*Hennig (1970)* suspected the distal basal article was an artifact and we can here confirm that the arista-like stylus of *Trichinites* consists of one basal article). Fore tibial gland absent or not visible; notum with four irregular rows of acrostichal and dorsocentral setulae; two pairs of larger dorsocentral setae; wing with costal vein ending beyond apex of $M_1$; Sc incomplete; $R_{4+5}$ not forked; cell dm with veins $M_1$, $M_2$ and $M_4$ branching separately off apex of cell, reaching wing margin; cell cua moderately long, subequal to length of cell bm, truncate apically; CuA projecting slightly obliquely; CuA+CuP reaching wing margin but becoming evanescent; anal lobe large. Terminal segments of female telescoping, without acanthophorite spines; cerci long.

We compared the female of *T. cretaceus* with our specimens of *Grimaldipeza* n. gen. and noticed several differences: fore tibial gland present (absent or not visible in *Trichinites*), terminal abdominal segments telescoping but compact in comparison with *Trichinites*, tergite 8, syntergite 9+10 and cerci are half as long as in *Trichinites* (Fig. 8A). The wings also present some differences, including the length and apex of cell cua and the setae in the anal area are much larger in *Grimaldipeza* n. gen. than in *Trichinites*, principally in *Grimaldipeza* sp. 2. Accordingly, the specimens described herein could not be classified within *Trichinites*.

The most distinctive features of this new fossil genus are: three veins emitted from cell dm; unbranched $R_{4+5}$; truncate cell cua which is subequal in length with cell bm, both cells being apically aligned; and presence of the fore tibial gland. Furthermore, male specimens have symmetrical terminalia, which appears to be at most only slightly rotated. The presence of the fore tibial gland immediately assigns the genus to the Hybotidae, along with the slender apical sensillum on the stylus. The genus is excluded from the subfamilies Hybotinae and Tachydromiinae by the presence of cell dm emitting three veins. The long cell cua and the linear alignment with cell bm, excludes it from the Ocydromiinae, but it is most similar to the genera *Bicellaria* Macquart, 1823 (Bicellariinae) and *Trichinomyia* Tuomikoski, 1959 (Trichininae) (*Chvála, 1983*; figs 204, 206). These two genera also have similar nearly symmetrical male terminalia. The Bicellariinae is a distinct monophyletic lineage, defined on the basis of the loss of cell dm and the branches of M evanescent near mid wing, whereas the Trichininae is defined on the basis of symplesiomorphies (*Sinclair & Cumming, 2006*): dichoptic females, antennal stylus about half as long or shorter than the postpedicel, cell dm emitting three veins, proboscis short and directed downwards, ventral apodeme and postgonites absent. The subfamily contains only two genera, *Trichina* Meigen, 1830 and *Trichinomyia*. However, the position of Trichininae within Hybotidae remains unresolved. *Grimaldipeza* spp. remain apart from this lineage on the basis of the elongate mouthparts and antennal stylus longer than the postpedicel. Furthermore, *Grimaldipeza* n. gen. can be separated from *Trichinomyia* by the thoracic hairs and slender bristles, and wings without pterostigma. The male terminalia of *Trichina* is somewhat similar to *Grimaldipeza coelica* gen. et sp. n., however the hypandrium in the latter seems to be symmetrically bilobed. Wing venation and mouthparts are similar to Oedaleinae; however, our genus can be excluded from the subfamily because of the long, apical antennal stylus, which is usually greatly shortened (shorter than postpedicel) in Oedaleinae.

The genus *Apterodromia* Oldroyd, 1949 was transferred from the Tachydromiinae to the tribe Ocydromiini (now subfamily Ocydromiinae) (*Sinclair & Cumming, 2000*) and excluded from Trichininae with the argument that an elongated cell cua represents the ground plan condition of the hybotid lineage and the male terminalia characters should be used. The male hypopygium of *Grimaldipeza coelica* gen. et sp. n., as mentioned above, is symmetrical, not rotated, yet the genus *Trichinomyia*, classified in Trichininae by *Sinclair & Cumming (2006)* is hypothesized to be the sister group to the remaining Hybotidae on the basis of its symmetrical male hypopygium. Thus, we can assume that symmetrical male hypopygium and the absence of genital rotation in *Ecommocydromia difficilis* and

*Grimaldipeza coelica* gen. et sp. n. are primitive conditions. A similar condition has recently been described in the Ocydromiinae genus *Pseudoscelolabes* Collin, 1933 (*Barros et al., 2022*).

The phylogenetic position of *Ecommocydromia difficilis* remains uncertain in part mostly because several important characters are not visible, *e.g.*, apex of the arista-like stylus, basal portion of the wings, and mouthparts. However, the wing venation possibly indicates a close relationship with *Grimaldipeza* n. gen. and *Trichinites*.

## Ecology

The eyes of *Grimaldipeza* n. gen. are holoptic in males and dichoptic in females; this condition indicates that males probably formed aerial mating swarms (*Chvála, 1976*). The form of the females' mouthparts differs from that of males. This could be because of a natural dimorphism, common in Empidoidea, where the females have more prominent mouthparts than the males (*Bletchly, 1954*). We cannot exclude that the females belong to different species because they are found in separate amber pieces. Consequently, they are not described here as new species. However, the morphology of the mouthparts indicates the well-known predatory feeding habits of the empidoids. Diptera are very abundant in Defaunation resin, copal, and amber, and Empidoidea are among the most abundant Diptera within Cretaceous ambers (*e.g.*, *Grimaldi & Cumming, 1999*; *Sinclair & Grimaldi, 2020*; *Ngô-Muller et al., 2021*). We know that selected taxa trapped in resins represent the fauna living in and around the resin-producing tree and appear in resins because of their ecology and behavior (*Solórzano-Kraemer et al., 2018*). In the case of the herein described specimens, their capture in resin is most probably due to swarming and predatory behaviors (*Chvála, 1976*; *Daugeron, 1997*).

The presence of *Grimaldipeza* n. gen. in the amber-bearing outcrops of El Soplao and La Hoya points out to a wide distribution of the genus in the Cretaceous Iberia Island, along the northern and eastern coasts. This kind of distribution is compatible with that of other taxa found in amber from the Maestrazgo Basin (such as San Just) and in amber from the Basque-Cantabrian Basin (such as El Soplao and Peñacerrada I). Species of Psocodea, Coleoptera, Hymenoptera, and Diptera have been found in San Just and amber-bearing outcrops of northern Iberia (*Arillo, Peñalver & Delclòs, 2008*; *Ortega-Blanco, Delclòs & Engel, 2011*; *Ortega-Blanco et al., 2011*; *Peris et al., 2014*; *Álvarez-Parra et al., 2022*). Thus, these paleogeographical distributions may indicate that the resiniferous forests in the Iberia Island were at least partly connected, not independently isolated, allowing the movement of entomofauna along the coastal forest environments. Furthermore, the finding of *Grimaldipeza* n. gen. in the El Soplao amber middle Albian in age, and La Hoya amber most probably lower Cenomanian in age, shows that this genus inhabited the Iberia Island for an interval of about seven million years (~107–100 Ma). The finding of nearly rounded kidney-shaped amber pieces in the same level together with aerial amber pieces in La Hoya amber-bearing outcrop implies a parautochthonous accumulation in a transitional environment (*Álvarez-Parra et al., 2021*), similarly to El Soplao (*Najarro et al., 2010*). The La Hoya FTIR spectrum (Fig. 10) does not show significant differences with the spectra of other ambers from the Maestrazgo Basin, such as San Just and Ariño

(*Álvarez-Parra et al., 2021*). The molecular composition of the amber from La Hoya, San Just, and Ariño relates the resin-producing tree to the Araucariaceae (*Menor-Salván et al., 2016*; *Álvarez-Parra et al., 2021*). Thus, the paleoenvironment could be similar in the three areas (E. Barrón, 2022, personal communication). Interestingly, the geochemical analysis of the El Soplao amber linked it to a resin-producing tree related to Cupressaceae or the extinct Cheirolepidiaceae, maybe the genus *Frenelopsis* (*Menor-Salván et al., 2010*). Therefore, the genus *Grimaldipeza* n. gen. could inhabit in forests with different plant compositions.

## CONCLUSIONS

The relevance of the new findings consists in providing new characters to the fossil character-pool of Hybotidae. Furthermore, a new genus and species are described, adding to the diversity of the family during the Cretaceous. This is critical in understanding the evolution of the family. The positions of *Grimaldipeza* n. gen., *Trichinites*, and *Ecommocydromia* remain unresolved until the extant subfamilies are better defined, principally the Trichininae, and more specimens in Cretaceous amber are discovered that could provide more key information. It is not possible to infer subfamily assignment with the information here recovered. Because Empidoidea, especially Hybotidae are frequent in amber, it is probably only a matter of time before new findings are discovered. Furthermore, delving into taphonomical, geochemical, and paleobotanical data of the amber-bearing outcrops in which these insects are found provide key information about their paleoenvironment and paleoecology.

The search for new characters in the fauna included in amber is supported by technologies such as μ-CT or SRμ-CT. However, not all the inclusions in the different ambers offer good results. The different contrasts of amber specimens are probably a matter of preservation, not only of the diagenesis of the amber itself but also of the diagenesis of the organism. In the case of the amber studied here, SRμ-CT, which normally offers a better contrast than μ-CT, did not provide any signal. This made the segmentation and therefore the visualization of the specimens impossible, thus the character search was limited to light microscopy.

## ACKNOWLEDGEMENTS

We would like to thank the colleagues who participated in fieldwork at the El Soplao and La Hoya amber outcrops. We also thank Ascensión Jarque, owner of the land in which La Hoya outcrop is located, and Federico Alegre and Vicente Arnau, discoverers of this amber outcrop. We are indebted to Rafael López del Valle for preparation of the amber pieces. We are grateful to Robin Kunz (Senckenberg Research Institute) for helping with the digitalization of the drawings, and to Eduardo Barrón (Instituto Geológico y Minero de España, CSIC) for providing stratigraphic and dating information of the amber outcrops. Thanks also to Thomas Schlüter (University of Swaziland) for providing valuable information about *Ecommocydromia difficilis*, to Andreas Abele-Rassuly and Christian Neumann (Museum für Naturkunde Berlin) for providing the holotype of *Ecommocydromia difficilis*, and Michael W. Rasser (Staatliches Museum für Naturkunde

Stuttgart) for providing the holotype of *Trichinites cretaceus*. The coauthor S.Á-P. thanks the support of his supervisors Xavier Delclòs (Universitat de Barcelona) and Enrique Peñalver (Instituto Geológico y Minero de España, CSIC). Agnieszka Soszyńska-Maj (University of Lodz), an anonymous reviewer, and the editor Kenneth De Baets (University of Warsaw) kindly commented on an earlier draft.

### Funding

This study is a contribution to the Spanish Ministry of Science and Universities (project AEI/FEDER, UE CGL2017-84419). This work was supported by the Consejería de Industria, Turismo, Innovación, Transporte y Comercio of the Gobierno de Cantabria through the public enterprise EL SOPLAO S.L. (research agreement #20963 with University of Barcelona, for the period 2022–2025). Sergio Álvarez-Parra was also supported by the Secretaria d'Universitats i Recerca de la Generalitat de Catalunya (Spain) and the European Social Fund (2021FI_B2 00003). Mónica M. Solórzano-Kraemer was supported by the Deutsche Forschungsgemeinschaft (DFG) (project SO 894/6-1). The funders had no role in study design, data collection and analysis, decision to publish, or preparation of the manuscript.

### Grant Disclosures

The following grant information was disclosed by the authors:
Spanish Ministry of Science and Universities: AEI/FEDER, UE CGL2017-84419.
Consellería de Industria, Turismo, Innovación, Transporte y Comercio of the Gobierno de Cantabria through the public enterprise EL SOPLAO S.L: #20963.
Secretaria d'Universitats i Recerca de la Generalitat de Catalunya (Spain).
European Social Fund: 2021FI_B2 00003.
Deutsche Forschungsgemeinschaft (DFG): SO 894/6-1.

### Competing Interests

The authors declare that they have no competing interests.

### Author Contributions

- Mónica M. Solórzano-Kraemer conceived and designed the experiments, performed the experiments, analyzed the data, prepared figures and/or tables, authored or reviewed drafts of the article, and approved the final draft.
- Bradley J. Sinclair analyzed the data, authored or reviewed drafts of the article, and approved the final draft.
- Antonio Arillo performed the experiments, analyzed the data, prepared figures and/or tables, authored or reviewed drafts of the article, and approved the final draft.
- Sergio Álvarez-Parra conceived and designed the experiments, performed the experiments, analyzed the data, prepared figures and/or tables, authored or reviewed drafts of the article, and approved the final draft.

## Data Availability

All data needed to evaluate the conclusions in the *paper* are presented in the table 1, in the figures 1 to 10, and in the Materials and Methods section. The specimens CES.404.1 ♂, CES.404.2 ♂, CES.439 ♂, and CES.372 ♀; are housed at the Colección Institucional del Laboratorio de la Cueva El Soplao in Celis, Cantabria (acronym for the collections is CES). Specimens MGUV-16348 and MGUV-16349 ♀ are housed at the Museu de la Universitat de València d'Història Natural (Burjassot, Valencia Province, Spain) (acronym for the collections is MGUV).

## New Species Registration

The following information was supplied regarding the registration of a newly described species:

Publication LSID: urn:lsid:zoobank.org:pub:D36ECF93-C05E-4A9C-8AA2-C799ED04346D

*Grimaldipeza* n. gen. LSID: urn:lsid:zoobank.org:act:23818B66-56D4-4D7B-BFA5-1EB7D6995462

*Grimaldipeza coelica* n. sp. LSID: urn:lsid:zoobank.org:act:9028DF70-F0B3-4A2C-B677-64F51C8CEC33.

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
