# Peer review of "A new genus of dance fly (Diptera: Empidoidea: Hybotidae) from Cretaceous Spanish ambers and introduction to the fossiliferous amber outcrop of La Hoya (Castellón Province, Spain)"

_PeerJ, doi:10.7717/peerj.14692_

## Round 0.1 · original submission · Minor Revisions

You describe various species of Cretaceous hybotids as well as a comprehensive and long overdue overview of the La Hoya amber deposit. The paper is well-written, cited literature well and appropriate chosen and easy to follow. There are however some crucial points I would like to see addressed in the revised version:

• Grimaldimyia versus Grimaldipeza: In the initial abstract, the specimens are still attributed to Grimaldimyia, which is very similar to Grimaldiomyia Lehrer. Such confusion should be avoided. I assume this issue will be resolved by using Grimaldipeza, but please check to be sure.

• Impact of the finds on our understanding of the evolution of the group: The impact of finds on the early evolution of this group should be more explicitly discussed (reviewer 1)

• Figures and Tables: illustrations (e.g., Fig.2C, 4C and 7D) and drawings should be improved (compare reviewer 2). I agree with reviewer 2 that a more structured table with all reconstructable details of morphology and detailed drawings of antennae, wing, legs and copulatory apparatus of the male would be indispensable.

• Formatting/Language/Typographic issues: There some issues falling into these categories particularly in the introduction which need to be resolved (compare reviewers and annotated pdf by myself)

• Location of the outcrops should likely be moved to materials and methods (compare reviewer 2)

Please make sure to address all points also additional ones listed in annotated pdfs and reviews.

I look forward to receiving the revised manuscript.

Reviewer 1 ·

Basic reporting

This is an interesting paper on Cretaceous hybotids, including a discussion on the nature of the La Hoya amber deposit. A couple of other Cretaceous fossil hybotid species are redescribed. The implications of the knowledge of these fossils could be better explored, even if they are initial hypotheses to be better explored. Comments are made on minor issues along the mark-up copy.

Experimental design

No comments.

Validity of the findings

Inferences made based on the findings are well set.

Additional comments

The meaning of the findings, in terms of implications to the understanding of the initial evolution of hybotids, could be better developed.

Annotated reviews are not available for download in order to protect the identity of reviewers who chose to remain anonymous.

·

Basic reporting

The paper is written in correct language using correct scientific vocabulary. The state of knowledge and research problems are included. In several places in the introduction I would make corrections (marked on the file). The literature is very rich, well selected, basic taxonomic works are cited as well as recent works and all literature covering the topic. I have highlighted a few items that are not in the references. I see an inconsistency in the citation of older works. Some are cited others are not.
Structure of the paper correct and conforms to standards. I wonder if the location of outcrops should not be moved to materials and methods.
From my perspective, the biggest improvement should be in the illustrations part. I think Figs. 2C and 4C do not contribute anything. The photos show the inclusions better. On the other hand, I would expect a more structured table, tables that show all the reconstructable details of the morphology. Based on the best elements of the construction of the holotype and paratypes. There should be one solid figure with drawings of the antenne, wing, legs and copulatory apparatus of the male. There should be one male genitalia to the species. One but a decent one, well reconstruct and described, definitely bigger. The antenna in figure 7d are misleading. It should be a one good reconstruction of one antenna. In this drawing it looks like there are two big scapus on one.

Experimental design

nothing to add

Validity of the findings

The paper is a very valuable source of new information about a poorly understood Cretaceous amber from Spain and a new position. It is imperative that it should be published after revisions.

---

## Round 0.2 · accepted · Accept

Thank you for addressing the suggestions of the reviewers and myself. Through the revised text and figures, your manuscript and interpretations have become (even) easier to follow and broader relevance. I just noted one misspelling in the acknowledgments which can be resolved during the proofing phase (see annotated pdf). It has been a great pleasure to handle your manuscript and I look forward to seeing your work published.